# Simulation and Experiment of Compression Molding Behavior of Substrate Block Suitable for Mechanical Transplanting Based on Discrete Element Method (DEM)

**Jingjing Fu** [1] , **Zhichao Cui** [1], **Yongsheng Chen** [1], **Chunsong Guan** [1], **Mingjiang Chen** [1,*] **and Biao Ma** [2,*]

[1]   Nanjing Institute of Agricultural Mechanization, Ministry of Agriculture and Rural Affairs,
      Nanjing 210014, China; tutujing12@163.com (J.F.); cuizhichaoabc@163.com (Z.C.); cys003@sina.com (Y.C.);
      cs.guan@163.com (C.G.)
[2]   Graduate School, Chinese Academy of Agricultural Sciences, Beijing 100083, China
*     Correspondence: chenmingjiang@caas.cn (M.C.); mabiao@caas.cn (B.M.)

**Abstract:** The compression molding performance of a substrate block has a significant effect on the quality and stability of mechanical transplanting. The physical experiment and DEM simulation were combined to evaluate the compression molding behavior of substrate block in this study. A calibration procedure of DEM parameters of peat particles was proposed at first. Then, the above parameters were brought into the contact model of the compression system–particles, and the effect of the loading speed on the compression behavior of the peat substrate block was investigated. The compressive force–displacement curves of the simulated and measured tests were all contained in the initial linear stage and non-linear stiffing stage. The particle number of central sections was higher than side section, and the variable coefficient was greater at higher loading speed. The substrate blocks all expanded after demolding. The higher the loading speed, the greater the expansion in the height's direction, and the easier it was for cracks to be generated near the bottom. This study will provide a reference for the design of substrate block forming machines.

**Keywords:** substrate block; compression molding; DEM; parameter calibration; seedling transplanting

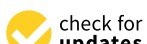

## 1. Introduction

Seedling transplanting is an important process in vegetable and economic crop production, and mechanical automation and intelligent production is an imperative portion of its development trend. However, at present, there still exists some problems, such as the rupture and fall of the seedling substrate or the damage of seedling stems and leaves, which will affect the quality, efficiency, and reliability of the transplantation [1]. Except for the structural design and stability of the transplanting machine, the quality of seedlings also has a significant effect on the seedlings' transplanting performance. A strong and unified seedling is expected to reduce management costs, and it is also suitable for mechanical operation. There are three types of seedlings for transplanting: bare root, plug trays, and pot/substrate block seedlings. The latter two are most used for mechanized transplantation [2,3]. The substrate block seedlings have better functional and developed root systems that are capable of withdrawing more water and nutrients, and they have higher survival rates than bare seedlings and plug seedlings [4,5]. Additionally, they also have relatively regular characteristics for the automatic mechanized transplanting than pot seedlings, which would help to expand the fully automated transplanting [6].

Although the substrate block is suitable for seedlings and mechanical operation, it also has stricter requirements than other types due to its compressed structure. In addition to the chemical properties (pH, EC, nutrients, etc.) provided by the components of the substrate materials, the physical characteristics have put forward higher requirements for the compressed substrate block, such as its mechanical strength, density, porosity,

water availability, air capacity and expansion rate [7]. Those are mainly dependent on the compression process. Some studies have investigated the moisture content of the raw materials, component formula, and compression parameters (compression stress, temperature) on the quality of the substrate block and the seedling's performance [8–10]. In addition to the above factors, the processing machinery of the substrate block is also one of the key factors affecting its performance. However, there is currently no certain standard for the compression molding of the substrate blocks, and there is no uniform manufacturing standard for molding equipment. Nowadays, the existing equipment used in China still has some problems, such as substrate shedding or sticking to the mold. Hence, studying the working mechanism between the substrate and molding machinery will help to improve the operation performance of the processing equipment and further improve the quality of substrate blocks. The force and movement process between the machine and substrate particles are complex. The combination of a simulation method and a practical experimental method is usually the commonly used and effective method. In the research on agricultural products and equipment, the discrete element method (DEM) is widely employed to capture the dynamic behavior of the particle materials and the interaction mechanisms between the materials and tools [11–14]. For the substrate block seedlings, understanding the microscopic processes of the particle flow and compression is critical for the quality control of the substrate block. The application of DEM can be beneficial to reduce the risk of failure and cost of production.

In this work, the commonly used peat was used as the growth substrate. The calibration of the DEM parameters of the peat was determined by a combination of a physical test and a simulation test. The stacking angle was adopted as a response value, and the optimal parameters were obtained by Plackett–Burman test, the steepest climbing test, and Box–Behnken test methods. Then, those calibrated parameters were applied into the DEM simulation of the compression molding of the substrate block. The effects of the loading speed on the compression behavior of the substrate block were evaluated from the screen of the distribution movement of the pressed particles. Finally, the reliability and accuracy of the simulation parameters were verified through experiments. It was expected to provide a theoretical basis for the simulation of experiment research and the preparation of substrate blocks with good qualities that are suitable for mechanical transplanting. It also provided an improvement direction for the design of a substrate block forming machine.

## 2. Materials and Methods

### 2.1. Determination of the Intrinsic Parameters of the Peat

2.1.1. Moisture Content and Density

The moisture content of peat was measured by the low-temperature drying method specified in GB/T 16913-2008. The density of the peat was determined using the pycnometer method according to GB/T 16913-2008. After calculation, the moisture content and density of the peat substate were 26.69% and 1560 kg·m$^{-3}$, respectively.

2.1.2. Stacking Angle

The injection limited bottom surface method, according to GB/T 11986-1989, was used to measure the stacking angle of the peat. The measuring device was, mainly, composed of a fixed frame, a funnel, and a steel plate. The front-view image of the peat pile was captured by a high-definition camera, and computer image processing technology was used to process the images. The software MATLAB was used to extract the image boundary pixels through edge detection, and the software Origin was used to pick up the boundary points and was further subjected to polynomial fitting. The arc tangent of the slope of the tangent line of the fitted curve was the stacking angle. The measured value of the stacking angle of the peat was 51.15°.

*2.2. Calibration of Discrete Element Simulation Parameters of the Peat*

In order to establish a proper model for simulating the compressing process of peat substrate blocks by the DEM method, the parameter calibration of the peat was the prerequisite. The related simulation parameters mainly include the intrinsic parameters of the peat and the contact parameters between the peat and peat as well as between the peat and boundary material (impact recovery coefficient, static friction coefficient, rolling friction coefficient, etc.). The stacking angle referred to the maximum slope angle accumulated and formed on the horizontal surface when particles slowly fell from a certain height under the action of gravity. It was usually taken as the response value, which was mainly influenced by the friction mechanisms between the particles and between the boundaries. Hence, the stacking angle test was performed and reproduced by numerical simulation to determine the optimal parameters of the DEM model.

### 2.2.1. Setting of the Simulations

The 3D model of the test device for the simulation stacking angle of the peat substate was established in the DEM software (EDEM 2020, Altair Engineering, Troy, MI, USA) according to its practical dimension. The material of the test device was set to stainless steel with a Poisson's ratio of 0.3, density of 7850 kg·m$^{-3}$, and shear modulus of $7 \times 10^{10}$ Pa. The single spherical particles with a radius of 1 mm with a normal distribution were used to form the peat particles. The Hertz–Mindlin model and the JKR contact model were adopted in this simulation.

### 2.2.2. The Sensitivity of the DEM Parameters to the Stacking Angle

In order to explore the sensitivity of the external input DEM parameters on the stacking angle, a Plackett–Burman (PB) test with 11 factors and 1 central point was designed using the software Design-Expert version 12 (Stat-Ease, Inc., Minneapolis, MN, USA). There were 9 real variables (A to I), and 2 other virtual variables (J and K) were used to estimate the errors, while each parameter was set to low and high levels, according to the recommended range value. According to related studies [15–18] and a large number of pre-tests in the early stage, the ranged of each parameter were selected, as listed in Table 1.

**Table 1.** The parameter table of the Plackett–Burman test.

| Symbol | Parameters | Low Level (−1) | High Level (+1) |
|:---:|:---:|:---:|:---:|
| A | Poisson's ratio of the particles | 0.2 | 0.4 |
| B | Shear modulus of the particles (MPa) | 1 | 10 |
| C | Impact recovery coefficient of particle–particle | 0.2 | 0.6 |
| D | Static friction coefficient of particle–particle | 0.2 | 0.8 |
| E | Rolling friction coefficient of particle–particle | 0.1 | 0.5 |
| F | Impact recovery coefficient of particle–steel | 0.2 | 0.6 |
| G | Static friction coefficient of particle–steel | 0.2 | 0.6 |
| H | Rolling friction coefficient of particle–steel | 0.05 | 0.25 |
| J | Surface energy (J·m$^{-2}$) | 0.1 | 1 |
| K, L | Virtual parameter | −1 | 1 |

### 2.2.3. Calibration of DEM Parameters of Peat

Based on the significant parameters selected by the Plackett–Burman test, the steepest climbing test was designed to further narrow the range of significant parameters. In the simulation test, the non-significant parameter was set to the middle value of the range, and the significant parameter was gradually increased, according to the designed step

length. The relative error between the simulation and the physical measured value of the stacking angle was taken as the evaluation index. Based on the results of the steepest climbing test, the Box–Behnken test was designed by the response surface methodology (RSM). The significant parameters were set with high, medium, and low levels. The set of non-significant parameters was the same as the steepest climbing test.

### 2.3. DEM Simulation for the Compression Molding of Peat Substrate Blocks

According to the size requirements of substrate blocks, a cuboid compression system was set in the DEM simulation environment (Figure 1a), which consisted of a top punch, a square tube, and a bottom plate. The compression press was simulated to understand the quantitative relationship between the DEM parameter settings and the compression behavior of simulated particles. The DEM parameters of particles were set based on the obtained optimal parameters from the results of the calibration of the DEM parameters of the peat. Before the compression simulation, a pre-test of the particle generation was carried out first. According to the actual load capacity of the mold and the pre-test of the particle generation, the results showed that 16,500 particles could almost fill the container. Additionally, further compression molding simulations were all based on the model with generated particles. During the simulation of compression molding process, the top punch part started to move down to compress the substrate with a loading displacement of 60 mm. The different loading speeds at 50 mm·min$^{-1}$, 100 mm·min$^{-1}$, 200 mm·min$^{-1}$, and 500 mm·min$^{-1}$ were set in the simulation. The compressive force–displacement curves and particle numbers on the center section, quarter section, and inner wall (Figure 1b) were collected in the post-simulation process.

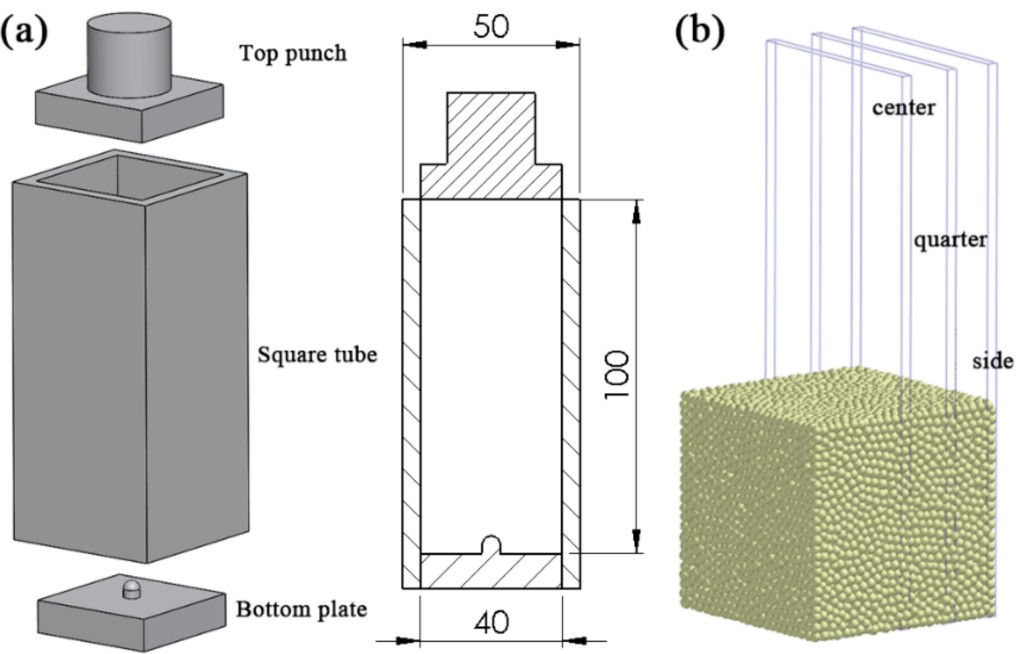

**Figure 1.** (**a**) Geometry of compression simulation platform and (**b**) data selection surfaces of particle numbers.

### 2.4. Experiment on the Compression Molding of Peat Substrate Blocks

The compression molding test of peat substrate blocks was carried out on a WDW-20 universal test machine (Jinan Chuanbai Instrument Equipment Co., Ltd., Jinan, China). The compression molding device consisted of a compression system and demolding part (Figure 2). The composition of the compression system was the same as the simulation platform, including a top punch, square tube, and bottom plate. When the compression molding test was carried out, the demolding part was not placed first. Before the test,

the square tube was filled in a natural loose state. The compression molding parameters were set similar to the simulation: a loading displacement of 60 mm and different loading speeds of 50 mm·min$^{-1}$, 100 mm·min$^{-1}$, 200 mm·min$^{-1}$, and 500 mm·min$^{-1}$. When the compression molding process was complete, the compression molding device was placed, as shown in Figure 2. Then, the peat substrate block was demolded by the top punch moving down.

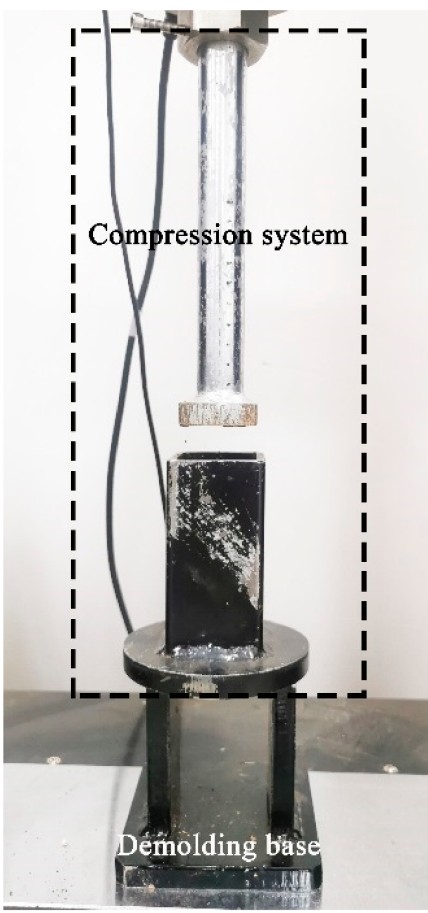

**Figure 2.** Compression molding device of peat substrate block.

### 3. Results and Discussion

*3.1. Calibration of the DEM Parameters of Peat*

3.1.1. Plackett–Burman (PB) Test

The protocol and results of the PB test and the significant results of each parameter are listed in Tables 2 and 3, respectively. The total contribution of B, D, and J was 83%, and they all significantly affected the stacking angle and followed the order: J > B > D. As shown in the Pareto chart (Figure 3), the contribution values of those three parameters were all exceeded the Bonferroni limit, which also indicated that the parameters were significant for the response. The contribution value lower than the t-value limit usually meant there was no effect for the response [19]. The contribution value of E between the Bonferroni limit and t-value limit was considered E and also had some effect on the response. Additionally, the information provided by Pareto chart showed that J and B had a positive effect on the response, while D and E had negative effects. Therefore, those four parameters were identified as the key parameters for the substrate to be calibrated in the steepest climbing test and BBD test below.

**Table 2.** The protocol and results of PB test.

| Run | A | B | C | D | E | F | G | H | J | K | L | Stacking Angle (°) |
|---|---|---|---|---|---|---|---|---|---|---|---|---|
| 1 | −1 | −1 | 1 | −1 | 1 | 1 | −1 | 1 | 1 | 1 | −1 | 59 |
| 2 | −1 | −1 | −1 | −1 | −1 | −1 | −1 | −1 | −1 | −1 | −1 | 32.76 |
| 3 | −1 | 1 | 1 | 1 | −1 | −1 | −1 | 1 | −1 | 1 | 1 | 41.43 |
| 4 | −1 | 1 | −1 | 1 | 1 | −1 | 1 | 1 | 1 | −1 | −1 | 57.17 |
| 5 | 1 | 1 | −1 | 1 | 1 | 1 | −1 | −1 | −1 | 1 | −1 | 33.38 |
| 6 | −1 | −1 | −1 | 1 | −1 | 1 | 1 | −1 | 1 | 1 | 1 | 46.23 |
| 7 | 1 | −1 | −1 | −1 | 1 | −1 | 1 | 1 | −1 | 1 | 1 | 37.47 |
| 8 | 0 | 0 | 0 | 0 | 0 | 0 | 0 | 0 | 0 | 0 | 0 | 61.64 |
| 9 | 1 | −1 | 1 | 1 | 1 | −1 | −1 | −1 | 1 | −1 | 1 | 44.28 |
| 10 | −1 | 1 | 1 | −1 | 1 | 1 | 1 | −1 | −1 | −1 | 1 | 54.66 |
| 11 | 1 | 1 | −1 | −1 | −1 | 1 | −1 | 1 | 1 | −1 | 1 | 84.23 |
| 12 | 1 | 1 | 1 | −1 | −1 | −1 | 1 | −1 | 1 | 1 | −1 | 83.65 |
| 13 | 1 | −1 | 1 | 1 | −1 | 1 | 1 | 1 | −1 | −1 | −1 | 43.13 |

**Table 3.** Significance analysis of parameters in PB test.

| p | Contribution (%) | *p*-Value | Order |
|---|---|---|---|
| A | 3.05 | 0.0822 | 6 |
| B | 21.02 | 0.0133 * | 2 |
| C | 3.05 | 0.0821 | 5 |
| D | 18.56 | 0.0150 * | 3 |
| E | 5.17 | 0.0509 | 4 |
| F | 1.42 | 0.1549 | 9 |
| G | 1.85 | 0.1253 | 8 |
| H | 1.88 | 0.1236 | 7 |
| J | 43.42 | 0.0065 ** | 1 |

Note: * and ** represent the significance at the 0.05 and 0.01 probability level, respectively.

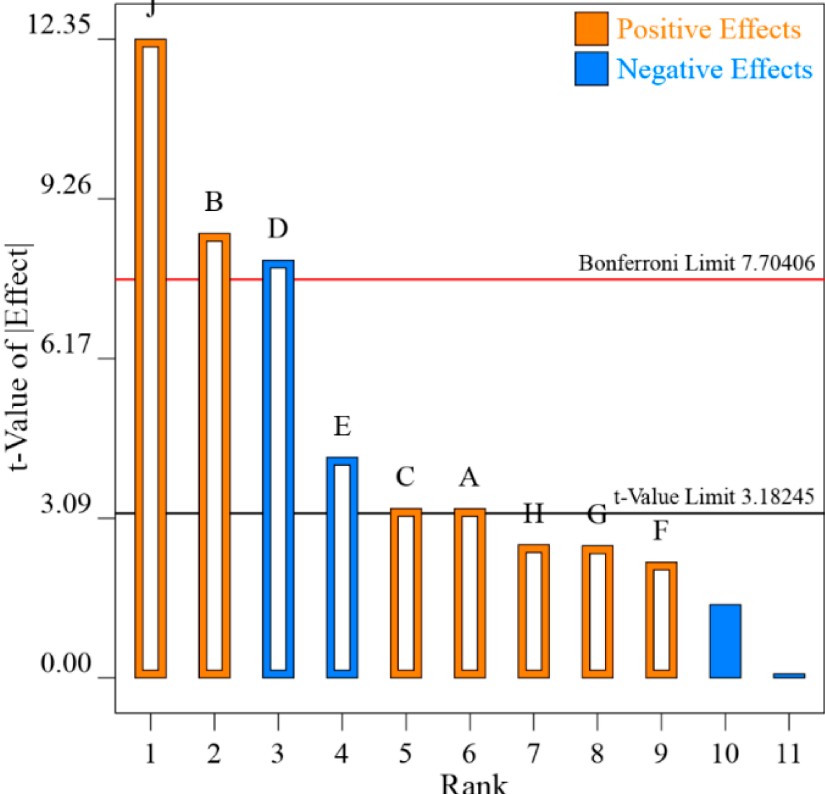

**Figure 3.** Pareto chart of parameters on response.

### 3.1.2. The Steepest Climbing Test

The design scheme and results of the steepest climbing test are presented in Table 4. The relative error was the value between the simulated stacking angle and the measured stacking angle (51.15°). The stacking angle gradually increased with the arrangement order of the test groups, and the relative error presented as first decreasing and then increasing. The stacking angle and relative error of the No.2 run were 51.07° and 0.16%, respectively. It indicated that the most optimal parameter level range was around the No.2 test, which was chosen to be the intermediate level (0) in the Box–Behnken test design. Additionally, the parameter levels of No.1 and No. 3 were chosen as the low (−1) and high levels (+1).

**Table 4.** Design scheme and results of steepest climbing test.

| No. | B | D | E | J | Stacking Angle (°) | Relative Error (%) |
|-----|------|------|-----|-------|--------------------|--------------------|
| 1 | 1 | 0.8 | 0.5 | 0.1 | 32.74 | 35.99 |
| 2 | 3.25 | 0.65 | 0.4 | 0.325 | 51.07 | 0.16 |
| 3 | 5.5 | 0.5 | 0.3 | 0.55 | 54.81 | 7.16 |
| 4 | 7.75 | 0.35 | 0.2 | 0.775 | 70.67 | 38.16 |
| 5 | 10 | 0.2 | 0.1 | 1 | 82.75 | 61.78 |

Note: positive effects (B and J) were in ascending order, and negative effects (D and E) were in descending order.

### 3.1.3. Box–Behnken Test

Based on the parameter selection from the PB test and the level selection from the steepest climbing test, a Box–Behnken test design (BBD) was carried out, and the results are shown in Table 5. The regression model describing the relationship between the stacking angle and selected parameters can be described as:

$$\text{Stacking angle} = 53.57 + 2.18B - 1.71D - 0.4179E + 6.81J + 1.25BD + 0.5006BE - 2.76BJ + 1.15DE - 1.01DJ + 1.53EJ + 0.0268B^2 - 0.6149D^2 + 2.86E^2 - 3.41J^2 \tag{1}$$

**Table 5.** BBD experiment results.

| Run | B | D | E | J | Stacking Angle (°) |
|-----|-----|-----|-----|-----|--------------------|
| 1 | 0 | 1 | −1 | 0 | 53.0542 |
| 2 | 0 | −1 | 1 | 0 | 56.04395 |
| 3 | 0 | 1 | 0 | 1 | 50.8817 |
| 4 | 1 | −1 | 0 | 0 | 55.8771 |
| 5 | 1 | 0 | −1 | 0 | 61.63415 |
| 6 | 0 | −1 | 0 | −1 | 46.189 |
| 7 | 0 | 0 | 0 | 0 | 53.98465 |
| 8 | 0 | 0 | 0 | 0 | 50.0464 |
| 9 | 0 | 0 | 0 | 0 | 53.55925 |
| 10 | 1 | 0 | 0 | −1 | 44.7601 |
| 11 | 0 | 0 | 0 | 0 | 53.65835 |
| 12 | 0 | 0 | −1 | −1 | 47.10265 |
| 13 | 0 | 1 | 0 | −1 | 41.7636 |
| 14 | −1 | 0 | 1 | 0 | 50.258 |
| 15 | −1 | 0 | 0 | −1 | 39.92395 |
| 16 | 0 | 0 | 1 | 1 | 62.2329 |
| 17 | 0 | −1 | 0 | 1 | 59.3303 |
| 18 | 0 | 0 | 1 | −1 | 44.90385 |
| 19 | 0 | 0 | −1 | 1 | 58.3123 |
| 20 | −1 | 0 | −1 | 0 | 55.354 |
| 21 | 1 | 0 | 1 | 0 | 58.5406 |
| 22 | −1 | 1 | 0 | 0 | 47.8322 |

**Table 5.** *Cont.*

| Run | B | D | E | J | Stacking Angle (°) |
|-----|-----|-----|-----|-----|--------------------|
| 23 | −1 | −1 | 0 | 0 | 51.86525 |
| 24 | −1 | 0 | 0 | 1 | 60.9074 |
| 25 | 1 | 1 | 0 | 0 | 56.82535 |
| 26 | 0 | −1 | −1 | 0 | 57.6126 |
| 27 | 1 | 0 | 0 | 1 | 54.70705 |
| 28 | 0 | 1 | 1 | 0 | 56.0758 |
| 29 | 0 | 0 | 0 | 0 | 56.5945 |

The ANOVA results of the quadratic model for BBD are listed in Table 6. The regression model's *p*-value = 0.0001 demonstrated a significant relationship between the responses and parameters. Consistent with the results of the PB test, B, D and J significantly affected the stacking angle. Moreover, the quadratic terms $E^2$ and $J^2$ had significant influence on stacking angle. The lack of fit was insignificant, and the correlation coefficient ($R^2 = 0.8962$) of the model was high, which demonstrated that the regression model fit well.

**Table 6.** ANOVA of the quadratic model for BBD.

| Source | Sum of Squares | df | Mean Square | F-Value | *p*-Value |
|--------|----------------|-----|-------------|---------|-----------|
| Model | 867.18 | 14 | 61.94 | 8.63 | 0.0001 ** |
| B | 57.22 | 1 | 57.22 | 7.98 | 0.0135 * |
| D | 34.97 | 1 | 34.97 | 4.87 | 0.0444 * |
| E | 2.10 | 1 | 2.10 | 0.2921 | 0.5974 |
| J | 556.63 | 1 | 556.63 | 77.58 | <0.0001 ** |
| BD | 6.20 | 1 | 6.20 | 0.8646 | 0.3682 |
| BE | 1.00 | 1 | 1.00 | 0.1397 | 0.7142 |
| BJ | 30.45 | 1 | 30.45 | 4.24 | 0.0585 |
| DE | 5.27 | 1 | 5.27 | 0.7342 | 0.4059 |
| DJ | 4.05 | 1 | 4.05 | 0.5640 | 0.4651 |
| EJ | 9.36 | 1 | 9.36 | 1.30 | 0.2725 |
| $B^2$ | 0.0047 | 1 | 0.0047 | 0.0007 | 0.9800 |
| $D^2$ | 2.45 | 1 | 2.45 | 0.3419 | 0.5680 |
| $E^2$ | 52.94 | 1 | 52.94 | 7.38 | 0.0167 * |
| $J^2$ | 75.29 | 1 | 75.29 | 10.49 | 0.0059 ** |
| Residual | 100.44 | 14 | 7.17 | | |
| Lack of fit | 78.70 | 10 | 7.87 | 1.45 | 0.3851 |
| Pure error | 21.74 | 4 | 5.44 | | |
| Cor Total | 967.62 | 28 | | | |

Note: * and ** represent the significance at the 0.05 and 0.01 probability level, respectively.

By considering the measured stacking angle (51.15°) as the target value and solving the regression model by using the optimization function of the software Design Expert, the optimal values of the four parameters were obtained as B = 2.43 MPa, D = 0.77, E = 0.41, and J = 0.34 J·m$^{-2}$. The obtained optimal parameters were employed to perform the DEM validation test. The simulated stacking angle was 50.89°, and its relative error with the measured stacking angle was 0.51% (Figure 4), indicating that the DEM models calibrated in this work were found to have good accuracy.

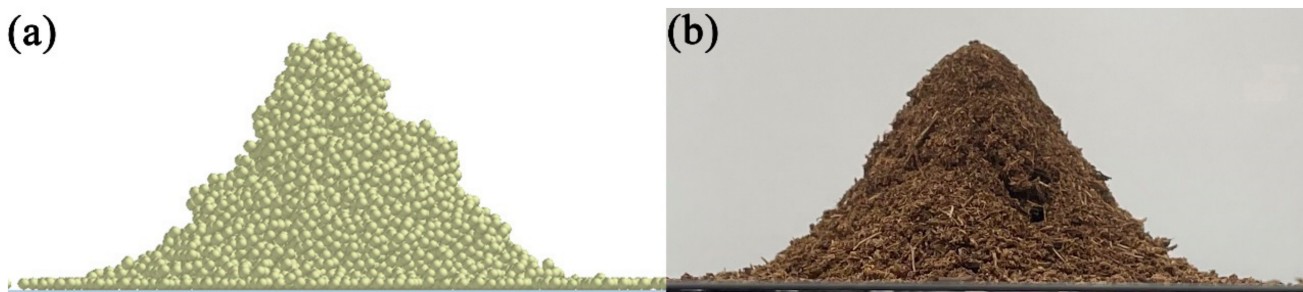

**Figure 4.** Comparison of the shape of peat pile between the (**a**) simulation test and (**b**) physical test.

### 3.2. Compression Molding Behavior of Substrate Block

The curves of the force–displacement responses in the compression molding process of the substrate block of measured and simulated experiments under different loading speeds are displayed in Figure 5a,b. The compressive force slightly increased when the displacement was lower than 40 mm and then increased rapidly until it reached the set displacement in both the measured and simulated processes. The first stage (displacement < 40 mm) could be considered as the linear compression process, which was mainly due to the pore reduction caused by the discharge of air and moisture. The second non-linear stage was the result of realignment of substrate particles. In this stage, the pores between the particles were filled, and their contact areas and frictional forces were increased, leading to further compression difficulties and jumpy pressures. This compression behavior was similar with other researchers, including an initial linear stage followed by a non-linear stiffening stage [20]. There was no great difference in the force–displacement curves among different loading speeds in either the measured or simulated processes. Therefore, only one set of the loading speeds (500 mm·min$^{-1}$) was selected for comparison (Figure 5c). It can be seen that the compression force began to increase after being compressed to 20 mm in the simulated process, while the force increased from the beginning of the compression in the measured process. It was due to the step of particle generation in the simulation. Before compression, the generated particles happened with a certain compression stack under their own gravity. Hence, in the beginning of the compression, the contact force between the top punch and particles was very small, and the two did not even touch. When pressing through this distance, the force of the simulation process increased significantly, and it, essentially, matched the actual process when the punch reached 50 mm. The comparison of the maximum compressive force of the measured and simulated tests is exhibited in Table 7. The maximum compressive force increased with the increasing loading speed both in the measurement and simulation. At higher loading speeds, the compressed displacement of the substrate particles per unit of time was larger, and then they will receive a larger compressive force. The simulated results under different loading speeds were all higher than the measured results, which was due to the unavoidable experimental errors in practical tests. However, the relative error was lower than 15%, especially at a loading speed of 500 mm·min$^{-1}$ (5.45%), which demonstrated the DEM simulations had a good match with experimental measurements, and the DEM model could capture the compression behavior of particles.

The instant particle number at the set displacement and the total accumulated particle number of three selected sections are shown in Table 8 and Figure 6. When the top punch reached the instant of the set displacement, the center and quarter sections possessed similar particle numbers around 850, while those in the side section were only in the range of 650–690 at all different loading speed conditions. Additionally, the variable coefficient was larger at higher loading speeds. This phenomenon also happened in the total accumulated particle number, and the difference between the inside section (center and quarter) and side section was greater. The average and total accumulated particle numbers decreased with increasing loading speed. It was noted that when the displacement of the punch part exceeded 40 mm, there was a trend in the accumulated particle numbers

of the quarter section being larger than the center section (Figure 6). As mentioned above, the particles were subjected to the non-linear stage of particle realignment, and the center particles started to spread around, causing the increasing particle number of the quarter section. Hence, at the instant of the set displacement, the quarter section had more particles. However, in total, the center section possessed the most total accumulated particles.

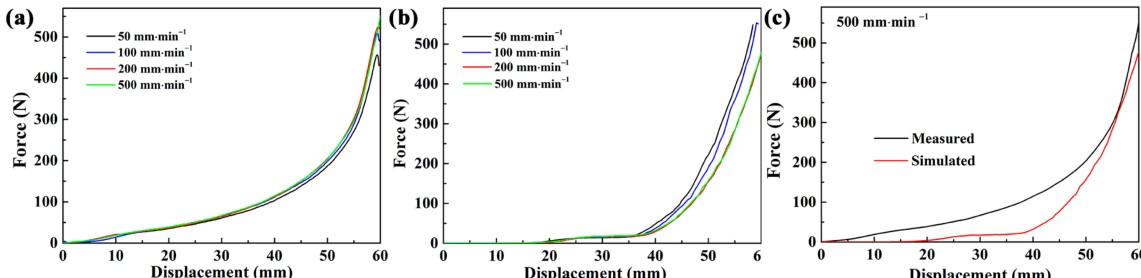

**Figure 5.** Compressive force–displacement curves of compression process: (**a**) measured, (**b**) simulated, and (**c**) comparison of measured and simulated process at 500 mm·min$^{-1}$ loading speed.

**Table 7.** The maximum compressive force of measured and simulated tests and their relative error.

| Loading Speed (mm·min$^{-1}$) | Maximum Compressive Force (N) | | Relative Error (%) |
|---|---|---|---|
| | Measured | Simulated | |
| 50 | 456.2 | 520.9 | 14.18 |
| 100 | 509.6 | 550.9 | 8.10 |
| 200 | 526.6 | 573.0 | 8.81 |
| 500 | 550.8 | 580.8 | 5.45 |

Note: relative error = |simulated − measured|/measured × 100%.

**Table 8.** The instant particle number and average particle number of selected sections at set displacement.

| Loading Speed (mm·min$^{-1}$) | Particle Number | | | Average Particle Number | Variable Coefficient (%) |
|---|---|---|---|---|---|
| | Center | Quarter | Side | | |
| 50 | 827 | 861 | 690 | 793 (90.52) | 11.42 |
| 100 | 836 | 848 | 691 | 792 (87.39) | 11.04 |
| 200 | 823 | 836 | 660 | 773 (98.08) | 12.69 |
| 500 | 814 | 841 | 655 | 770 (100.50) | 13.05 |

The heights of the peat substrate blocks of different loading speeds after standing for 48 h were collected: 63 (±1) mm, 64 (±1) mm, 65 (±1) mm, and 67 (±1) mm, respectively. According to the length of the tube and the moving displacement of the punch, the theoretical height of a block should be 40 mm, which indicates that the substrate blocks expanded in the height direction after compression molding. Additionally, their surfaces have cracks of varying degrees: as the loading speed increased, the location of the crack was closer to the bottom (Figure 7). At slower loading speeds, the firmness of the block from bottom to top was better, which could be reflected by the particle number. Therefore, the higher the loading speed the easier it was for the blocks to expand after molding to form cracks from the bottom. However, in order to achieve a certain production efficiency in actual production, the compression speed is, generally, fast. Therefore, in order to reduce the expansion problem of the substrate block and ensure its quality, it is recommended to explore the molding quality through some methods, such as adding agglomerants.

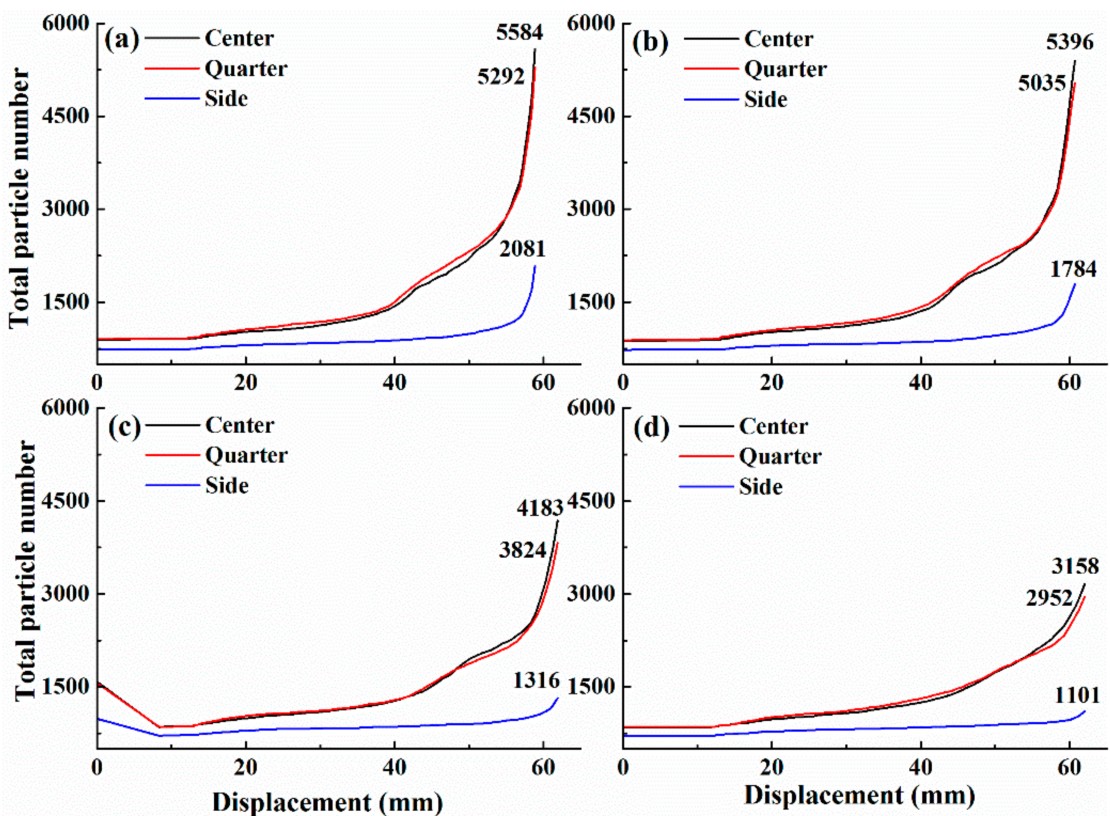

**Figure 6.** Total accumulated particle number on selected sections of substrate block at different loading speed: (**a**) 50 mm·min$^{-1}$, (**b**) 100 mm·min$^{-1}$, (**c**) 200 mm·min$^{-1}$, and (**d**) 500 mm·min$^{-1}$.

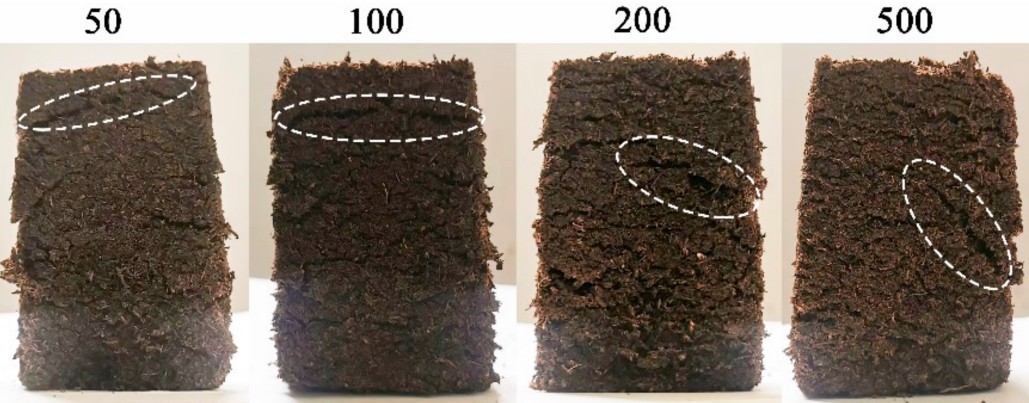

**Figure 7.** Image of peat substrate blocks with different loading speed after standing 48 h.

## 4. Conclusions

The compression molding behavior of peat substrate blocks was evaluated by the combination of a physical experiment and simulation. The simulation parameters of peat particles were calibrated first by the discrete element and response surface methodology. Then, the calibrated parameters were brought into the contact model of the compression system–particles, and the effect of the loading speed on the compression behavior of the substate blocks was investigated. The following conclusions were drawn:

(1)  In the proposed calibration method of the DEM parameters of peat particles, the shear modulus of the particles, the static friction coefficient of the particle–particle interactions, the rolling friction coefficient of the particle-particle interactions, and surface energy were found have significant effects on the stacking angle of peat through the Plackett–Burman test. After the range was narrowed by the steepest climbing

test, the significant parameters were optimized by the second-order regression model established in the Box–Behnken test. The result of the validation test of the stacking angle of peat under optimal parameters showed the average relative error between the simulated (50.89°) and the physical stacking angle (51.15°) was 0.51%, indicating the good reliability and accuracy of the DEM model and the simulated parameters of the peat particles.

(2) In the physical and simulated experiments of the compression molding of peat substrate blocks, the compression force of the measured and simulated test all included an initial linear stage followed by a second non-linear stiffening stage in the whole compression process. Additionally, the maximum force increased with increasing loading speed in both the measured and simulated test. The relative errors between measured and simulated maximum force were around 10%, and, especially, the loading speed at 500 mm·min$^{-1}$ was 5.45%. In general, the DEM simulations had a good match with the experimental measurements, and the DEM model could capture the compression behavior of particles.

(3) The peat substrate blocks all exhibited surface cracks under different loading speeds, and the location of the crack was closer to the bottom as the loading speed increased. It was due to the inhomogeneous particle distribution, resulting in varying degrees of compaction along the forming direction at higher loading speeds. While the practical production of substrate blocks mostly involves rapid prototyping to ensure operational efficiency, the compression molding of pure peat usually has lower mechanical properties that do not meet the requirements of mechanical transplanting. Hence, it is suggested to add some agglomerant to enhance the interparticle binding.

In summary, this study established a DEM model that could better simulate the compression molding of peat particles to prepare substrate blocks, which will contribute to the performance improvements of substrate blocks. The finding of this work could provide a reference for the design of the forming machine of substrate blocks suitable for mechanical transplanting.

**Author Contributions:** Conceptualization, J.F., Y.C. and M.C.; methodology, J.F., Z.C., C.G. and B.M.; software, J.F.; validation, J.F., M.C. and B.M.; formal analysis, J.F.; investigation, J.F., Z.C. and C.G.; data curation, J.F.; writing—original draft preparation, J.F.; writing—review and editing, J.F., M.C. and B.M.; visualization, J.F.; supervision, Y.C. and M.C.; funding acquisition, C.G., Z.C. and Y.C. All authors have read and agreed to the published version of the manuscript.

**Funding:** This research was funded by the Nanjing Modern Agricultural Machinery Equipment and Technology Innovation Demonstration Project (Grant No. NJ [2022] 03), Jiangsu Agricultural Science and Technology Innovation Fund (Grant No. CX (22) 3092), and Innovative research group of Agricultural Production Waste Resource Utilization Equipment.

**Institutional Review Board Statement:** Not applicable.

**Data Availability Statement:** Data are contained within the article.

**Acknowledgments:** The authors thank the editor and anonymous reviewers for providing helpful suggestions for improving the quality of this manuscript.

**Conflicts of Interest:** The authors declare no conflict of interest.

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
