# Peer review of "Simulation and Experiment of Compression Molding Behavior of Substrate Block Suitable for Mechanical Transplanting Based on Discrete Element Method (DEM)"

_agriculture, doi:10.3390/agriculture13040883_

Round 1

Reviewer 1 Report

In this paper, the main influencing factors of peat particle stacking angle are determined by PB test, and the significant parameters are optimized by BB test. The parameters of peat matrix simulation model are calibrated, and the reliability of the model is verified.The mechanical behavior of matrix block was studied by changing the loading speed.The method of the paper is proper and the thinking is clear. However, the following contents still need to be supplemented and revised:

1.The object of mechanical transplanting is seedlings with substate blocks, and the mechanical properties of substate blocks will also be affected by the roots of seedlings. Therefore, it is more practical to study the mechanical properties of substate blocks under the combined action of roots and soil.

2. The paper has completed the simulation and physical experiments, but it lacks the support of the experimental data of the real transplanting scene.

3.Line138-139, what is the basis for selecting the setting value of loading speed?

Author Response

In this paper, the main influencing factors of peat particle stacking angle are determined by PB test, and the significant parameters are optimized by BB test. The parameters of peat matrix simulation model are calibrated, and the reliability of the model is verified. The mechanical behavior of matrix block was studied by changing the loading speed. The method of the paper is proper and the thinking is clear. However, the following contents still need to be supplemented and revised:

1.The object of mechanical transplanting is seedlings with substate blocks, and the mechanical properties of substate blocks will also be affected by the roots of seedlings. Therefore, it is more practical to study the mechanical properties of substate blocks under the combined action of roots and soil.

  • Response: Thanks for the good advice. As you mentioned, except for the compression molding parameters, the roots of seeding also have effect on the mechanical properties of substrate block. In this work, our objective was to evaluate the compression molding behavior of substrate block with a certain strength so that it could be applied in the mechanical transplanting. This work is the first step, our further work will study the performance of substrate block under combined action of roots and soil.

  1. The paper has completed the simulation and physical experiments, but it lacks the support of the experimental data of the real transplanting scene.
  • Response: Thanks for the good advice. The precondition of the mechanical transplanting is that the substrate block must have better strength and seedling effect. This work mainly focused on the compression molding of substrate block. As your suggestion, the seedling effect and transplanting performance will be studied in future.

3.Line138-139, what is the basis for selecting the setting value of loading speed?

  • Response: The selecting basis of the loading speed was set according to the range of permissible conditions of the universal testing machine. The loading speed range of the universal testing machine used in this work is 0.01-500 mm·min-1. In order to ensure a certain production efficiency, a larger speed was selected for testing.

Reviewer 2 Report

The manuscript “Evaluation of the compression molding behavior of substate block suitable for mechanical transplanting based on experiment and DEM simulation” by Jingjing Fu et al. presents physical experiments and DEM simulations. This manuscript represents a contribution to your valuable Journal.

However, before the editor makes a decision, I suggest that the authors must take into account the following corrections:

1. The English, while generally good, has minor errors. Therefore, thorough proofreading is needed.

2. The author should clearly explain the novelty and what the added value of this article is in the abstract, and more references need to be reviewed.

3. Conclusion should be more carefully rewritten, summarizing what has been learned and why it is exciting and valuable.

4. The introduction involve the development status of simulation. I am convinced that it would be helpful for the manuscript if the following papers could be included in the References section, for example:

• Coupled GIMP and CPDI material point method in modelling blast-induced three-dimensional rock fracture, International Journal of Mining Science and Technology 32 (5), 1097-1114.

Author Response

The manuscript “Evaluation of the compression molding behavior of substate block suitable for mechanical transplanting based on experiment and DEM simulation” by Jingjing Fu et al. presents physical experiments and DEM simulations. This manuscript represents a contribution to your valuable Journal.

However, before the editor makes a decision, I suggest that the authors must take into account the following corrections:

  1. The English, while generally good, has minor errors. Therefore, thorough proofreading is needed.
  • Response: The whole text has been carefully read and checked, some English writing has been revised. Please see the manuscript highlighted in red.

  1. The author should clearly explain the novelty and what the added value of this article is in the abstract, and more references need to be reviewed.
  • Response: Thanks for the good advice. The abstract and introduction have been revised. Please see the manuscript highlighted in red.

  1. Conclusion should be more carefully rewritten, summarizing what has been learned and why it is exciting and valuable.
  • Response: Thanks for the good advice. The conclusion has been carefully revised.

  1. The introduction involve the development status of simulation. I am convinced that it would be helpful for the manuscript if the following papers could be included in the References section, for example:
  • Coupled GIMP and CPDI material point method in modelling blast-induced three-dimensional rock fracture, International Journal of Mining Science and Technology 32 (5), 1097-1114.
  • Response: Thanks for the good advice. The following paper has been cited.

  1. Wan, D.; Wang, M.; Zhu, Z.; Wang, F.; Zhou, L.; Liu, R.; Gao, W.; Shu, Y.; Xiao, H. Coupled GIMP and CPDI material point method in modelling blast-induced three-dimensional rock fracture. International Journal of Mining Science and Technology 2022, 32, 1097-1114.

Reviewer 3 Report

Summary: This paper presents a hybrid experimental-numerical approach to study the compression behavior of substrate block for mechanical transplanting. The authors performed numerical simulation using DEM, which was validated using experiments. The paper is of interest and fits well within the scope of agriculture journal.

Comment(s):

1.Overall, the paper readability is poor. Lots of sentences should be re-written in a way to make it easier for the reader to understand. As an example, in the introduction, lines 28-29: “Except for the transplanting machine performance, the quality of seedlings also possesses a significant position”. What does this mean?

2.The authors stated that the moisture content of the peat is 26.69%. How was this accounted for in the simulations?   

3.It appears that the peat particles in the experiment were poured in the mold in its loosest state, in a process close to pluviation. How were the peat particles in the DEM generated? Was there any overlap when the initial assembly was generated? More discussion is needed on the initial assembly generation in the DEM, as the initial fabric highly impacts the final results.

4.What is the convergence criteria used in the DEM models? How is the quasi-static equilibrium ensured in the simulations?

5. Table 5. What is the title of it?

6. Figure 5: to say that the DEM model reproduces the experiment, the curves in Figure 5 a and b should be overlapped for each loading speed. Differences between the model and experiment at each loading speed should be discussed.

7. Assuming the DEM model and the experimental results match (as in Figure 5), how is this by itself helpful? What is the point of only running the DEM model and show it matches the experiments? More simulations should be performed to highlight the impact of each input parameter and give recommendations. Otherwise, there is no point of including the DEM results.  

8. Title of the paper has a type. Please correct it. 

Author Response

Summary: This paper presents a hybrid experimental-numerical approach to study the compression behavior of substrate block for mechanical transplanting. The authors performed numerical simulation using DEM, which was validated using experiments. The paper is of interest and fits well within the scope of agriculture journal.

Comment(s):

1.Overall, the paper readability is poor. Lots of sentences should be re-written in a way to make it easier for the reader to understand. As an example, in the introduction, lines 28-29: “Except for the transplanting machine performance, the quality of seedlings also possesses a significant position”. What does this mean?

  • Response: Thanks for the good advice. This sentence has been revised to “Except for the structural design and stability of transplanting machine, the quality of seedlings also has significant effect on the seedlings transplanting performance.”. And the English writing of the whole text has been carefully checked and revised. Please see the manuscript heighted in red.

2.The authors stated that the moisture content of the peat is 26.69%. How was this accounted for in the simulations?

  • Response: Thanks for the good advice. Peat with different moisture content has different density. At the moisture content of 26.69%, the density is 1560 kg·m-3. Only the density data of the particles is needed in the simulation.

3.It appears that the peat particles in the experiment were poured in the mold in its loosest state, in a process close to pluviation. How were the peat particles in the DEM generated? Was there any overlap when the initial assembly was generated? More discussion is needed on the initial assembly generation in the DEM, as the initial fabric highly impacts the final results.

  • Response: Thanks for the good advice. In fact, before the start of the simulation test, the mold is filled with particles under each test condition. The compression molding simulation is based on the model with generated particles. According to the carrying capacity of the actual mold, the pre-test of particle generation was carried out first. The results showed that 16,500 particles can almost fill the mold, which is similar to the actual situation. Hence, the further simulation directly started with compression. Those process has been modified and supplemented in detail in the experiment part highlighted in red.

4.What is the convergence criteria used in the DEM models? How is the quasi-static equilibrium ensured in the simulations?

  • Response: Thanks for the good advice. In my knowledge level, the DEM simulation of pure solid particles should have no convergence criteria. The particles will always have extremely small velocity values during the compression and stacking process, and cannot fully achieve static equilibrium. Convergence conditions exist in particle-fluid DEM simulation. It may also be that my theoretical knowledge level is not enough, and I don’t know the knowledge of convergence standards. I hope the reviewer can give me some suggestions.

  1. Table 5. What is the title of it?
  • Response: This is our carelessness, the title of Table 5 is “BBD experiment results”.

  1. Figure 5: to say that the DEM model reproduces the experiment, the curves in Figure 5 a and b should be overlapped for each loading speed. Differences between the model and experiment at each loading speed should be discussed.
  • Response: Thanks for the good advice. The fig.5 has been revised, and relevant discussion has been added. Details are in line “There was no great difference of the force-displacement curves among different loading speed no matter the measured or simulated process. Therefore, only one set of the loading speed (500 mm·min-1) was selected for the comparison (Fig. 5c). It can be seen that the compression force began to increase after being compressed to 20 mm in the simulated process, while in the measured process, the force increased from the begin-ning of the compression. It was due to the step of particle generation in the simulation. Before the compression, the generated particles have happened with a certain com-pression stack under their own gravity. Hence, in the beginning of the compression, the contact force between the top punch and particles was very small, even the two did not touch. When pressing through this distance, the force of the simulation process in-creased significantly, and it basically matched the actual process when the punch reached 50mm.”.

Figure 5. Compressive force-displacement curves of compression process: (a) measured, (b) simulated, and (c) comparison of measured and simulated process at 500 mm·min-1 loading speed.

  1. Assuming the DEM model and the experimental results match (as in Figure 5), how is this by itself helpful? What is the point of only running the DEM model and show it matches the experiments? More simulations should be performed to highlight the impact of each input parameter and give recommendations. Otherwise, there is no point of including the DEM results.
  • Response: Thanks for the good advice. Nowadays, there is no certain standard for the compression molding of the substrate block. The purpose of DEM simulation in this study is that if the process of particle compression can be simulated, it can provide a basis and reference for the design and parameter optimization of the molding machine according to the application requirements in the future. It also can reduce the amount of experiments and design costs.

  1. Title of the paper has a type. Please correct it.
  • Response: Thanks for the good advice. The title has been revised to “Simulation and Experiment of Compression Molding Behavior of Substate Block Suitable for Mechanical Transplanting based on Discrete Element Method (DEM)”.

Round 2

Reviewer 1 Report

Please further check the language expression and minor spelling mistakes.

Reviewer 2 Report

No more comments.